# Rutaecarpine Increases Nitric Oxide Synthesis via eNOS Phosphorylation by TRPV1-Dependent CaMKII and CaMKKβ/AMPK Signaling Pathway in Human Endothelial Cells

**DOI:** 10.3390/ijms22179407

**Published:** 2021-08-30

**Authors:** Gi Ho Lee, Chae Yeon Kim, Chuanfeng Zheng, Sun Woo Jin, Ji Yeon Kim, Seung Yeon Lee, Mi Yeon Kim, Eun Hee Han, Yong Pil Hwang, Hye Gwang Jeong

**Affiliations:** 1College of Pharmacy, Chungnam National University, Daejeon 34134, Korea; ghk1900@cnu.ac.kr (G.H.L.); chaeyeon05@o.cnu.ac.kr (C.Y.K.); phoenix-zcf@naver.com (C.Z.); mpassword@cnu.ac.kr (S.W.J.); jykim525@o.cnu.ac.kr (J.Y.K.); sy9842@o.cnu.ac.kr (S.Y.L.); kmymy@o.cnu.ac.kr (M.Y.K.); 2Drug & Disease Target Research Team, Division of Bioconvergence Analysis, Korea Basic Science Institute (KBSI), Cheongju 28119, Korea; heh4285@kbsi.re.kr; 3Fisheries Promotion Division, Mokpo 58613, Korea; protoplast@hanmail.net

**Keywords:** rutaecarpine, eNOS, AMPK, CaMKII, Ca^2+^, TRPV1

## Abstract

Rutaecarpine (RUT) is a bioactive alkaloid isolated from the fruit of *Evodia rutaecarpa* that exerts a cellular protective effect. However, its protective effects on endothelial cells and its mechanism of action are still unclear. In this study, we demonstrated the effects of RUT on nitric oxide (NO) synthesis via endothelial nitric oxide synthase (eNOS) phosphorylation in endothelial cells and the underlying molecular mechanisms. RUT treatment promoted NO generation by increasing eNOS phosphorylation. Additionally, RUT induced an increase in intracellular Ca^2+^ concentration and phosphorylation of Ca^2+^/calmodulin-dependent protein kinase kinase β (CaMKKβ), AMP-activated protein kinase (AMPK), and Ca^2+^/calmodulin-dependent kinase II (CaMKII). Inhibition of transient receptor potential vanilloid type 1 (TRPV1) attenuated RUT-induced intracellular Ca^2+^ concentration and phosphorylation of CaMKII, CaMKKβ, AMPK, and eNOS. Treatment with KN-62 (a CaMKII inhibitor), Compound C (an AMPK inhibitor), and STO-609 (a CaMKKβ inhibitor) suppressed RUT-induced eNOS phosphorylation and NO generation. Interestingly, RUT attenuated the expression of ICAM-1 and VCAM-1 induced by TNF-α and inhibited the inflammation-related NF-κB signaling pathway. Taken together, these results suggest that RUT promotes NO synthesis and eNOS phosphorylation via the Ca^2+^/CaMKII and CaM/CaMKKβ/AMPK signaling pathways through TRPV1. These findings provide evidence that RUT prevents endothelial dysfunction and benefit cardiovascular health.

## 1. Introduction

Endothelial activation maintains vascular homeostasis via the release of various vasoactive substances, including nitric oxide (NO) [1]. NO, a biomolecule that plays a physiologically protective role, is synthesized by endothelial NO synthase (eNOS) from the amino acid l-arginine in endothelial cells [2]. Furthermore, in the endothelium, NO exhibits a vascular protective effect through various actions such as inhibition of the adhesion and migration of immune cells, as well as aggregation and thrombosis of platelets [3]. Therefore, the production of NO through eNOS activation in vascular endothelium should be taken into consideration in studies regarding the treatment and prevention of vascular disease, including atherosclerosis.

eNOS activity is regulated by phosphorylation at several sites, including Ser1177 [4]. The phosphorylation of eNOS at Ser1177 is modulated by various kinases such as AMP-activated protein kinase (AMPK), Ca^2+^/calmodulin-dependent kinase II (CaMKII), and Ca^2+^/calmodulin-dependent protein kinase kinase β (CaMKKβ) [5,6,7]. Increases in intracellular Ca^2+^ upregulate the phosphorylation of calmodulin-dependent kinases through activation of calmodulin, consequently inducing the phosphorylation of eNOS and generating NO [8]. In Ca^2+^-dependent eNOS activation, AMPK, CaMKII, and CaMKKβ are activated by the Ca^2+^ signaling pathway and play important roles in closely regulating eNOS Ser1177 phosphorylation [9].

Transient receptor potential vanilloid type 1 (TRPV1), also known as the capsaicin receptor, is a member of the transient receptor potential family of ion channels [10]. Activation of TRPV1 by heat, protons, and vanilloids increases the level of intracellular Ca^2+^ and leads to induction of the phosphorylation of eNOS and NO production [11]. Furthermore, a recent study indicated that TRPV1 is involved in the protective effect in cardiovascular diseases by regulating vascular tone and endothelial function [12,13].

Rutaecarpine (RUT), a bioactive component isolated from *Evodia rutaecarpa*, has been used in traditional Chinese medicine to treat hypertension, headache, and gastrointestinal disorders and shows an anti-inflammatory and anti-thrombotic effect [14]. The protective effects of RUT can be blocked by TRPV1 antagonists, suggesting that RUT exerts a protective effect mainly through its agonistic effects on TRPV1 [15,16]. RUT was reported to inhibit intimal hyperplasia in a balloon-injured rat artery model and to suppress angiotensin II-induced rat vascular smooth muscle cell proliferation [17,18]. Despite the cardiovascular protective effect of RUT, its modulatory effect on NO synthesis through eNOS activation and its molecular mechanism of action remain unclear. Thus, we investigated the effects of RUT on NO synthesis via eNOS phosphorylation by the TRPV1-dependent Ca^2+^/AMPK signaling pathway in human endothelial EA. hy926 cells.

## 2. Results

### 2.1. Effects of RUT on the Cell Viability and Cytotoxicity in Endothelial Cells

RUT chemical structure is shown in Figure 1A. To determine the effects of RUT on cytotoxicity, MTT and LDH assays were performed by treating endothelial cells with various concentrations (1–20 μM) RUT for 24 h. RUT had no effect on cell viability or cytotoxicity at concentrations of 1 to 10 μM (Figure 1B,C). However, treatment of 20 μM with RUT showed significant effects on cell viability and cytotoxicity. Therefore, 10 μM was used as the maximum concentration of RUT in subsequent experiments.

### 2.2. Effects of RUT on eNOS Phosphorylation and NO Generation

The activity of eNOS is regulated by phosphorylation and is closely related to NO generation in human endothelial cells [19]. We investigated the effects of RUT on eNOS phosphorylation and NO generation in human endothelial cells. RUT increased the eNOS phosphorylation from 15 min, and the eNOS phosphorylation was induced from 1 μM RUT in endothelial cells (Figure 2A,B). Furthermore, treatment with RUT 10 μM significantly induced NO generation, whereas pretreatment with 100 μM L-N^G^-Nitro arginine methyl ester (L-NAME), an NOS inhibitor, attenuated on RUT-induced NO generation (Figure 2C). NO generation was detected using DAF-2DA, a marker of oxidized species of NO. These results indicate that RUT increased NO generation by inducing the phosphorylation of eNOS in human endothelial cells.

### 2.3. Effects of RUT on the Intracellular Ca^2+^ Levels and Phosphorylation of CaMKII, CaMKKβ and AMPK

Increased intracellular Ca^2+^ induces eNOS phosphorylation at Ser1177 through Calmodulin activation, leading to NO generation [5]. We assessed the intracellular Ca^2+^ levels to determine the role of the Ca^2+^ signaling pathway in the effect of RUT on eNOS phosphorylation in endothelial cells. As shown in Figure 3A, RUT treatment concentration-dependently increased intracellular Ca^2+^ levels, as determined using fluo-4AM, a dye that emits fluorescence as it binds intracellular Ca^2+^. Additionally, the levels of CaMKKβ, CaMKII, and AMPK phosphorylation were significantly increased by RUT in endothelial cells (Figure 3B,C). These results suggest that an increase in intracellular Ca^2+^ levels is required for RUT to increase NO production through phosphorylation of eNOS.

### 2.4. Effects of TRPV1 Blockade on Intracellular Ca^2+^ Levels and Phosphorylation of CaMKKβ, CaMKII, AMPK and eNOS in Response to RUT

Previous studies reported that transient receptor potential (TRP) channels locate on the plasma membrane in endothelial cells and prevent endothelial dysfunction by increasing the eNOS phosphorylation and NO synthesis [20,21]. A TRPV1-selective antagonist, SB366791, was used to determine whether TRPV1 is required for eNOS phosphorylation and activation of Ca^2+^-dependent kinase protein by RUT. As shown in Figure 4A, pretreatment with 10 μM SB366791, 10 μM BAPTA-AM (an intracellular calcium chelator), or 0.5 mM EDTA (an extracellular calcium chelator) inhibited the RUT-mediated increase in intracellular Ca^2+^ levels. In addition, pretreatment with 10 μM SB366791 and 10 μM W7 (a calmodulin antagonist) suppressed RUT-induced eNOS phosphorylation (Figure 4B). Furthermore, both antagonists inhibited RUT-induced phosphorylation of CaMKKβ, CaMKII, and AMPK (Figure 4C). These results indicate that TRPV1 activation leads to the phosphorylation of CaMKKβ, CaMKII, and AMPK, which is required for RUT-induced eNOS phosphorylation and NO generation.

### 2.5. Roles of CaMKII and the CaMKKβ/AMPK Signaling Pathway in RUT-Induced eNOS Activation and NO Generation

The activity of CaMKII and AMPK is related to the phosphorylation of eNOS at ser1177 in endothelial cells [22]. We used KN-62 (a CaMKII inhibitor), STO-609 (a CaMKKβ inhibitor), and Compound C (an AMPK inhibitor) to examine the mechanisms underlying eNOS phosphorylation and NO generation in response to RUT. Western blotting indicated that pretreatment with KN-62 suppressed RUT-induced eNOS phosphorylation in endothelial cells (Figure 5A). Additionally, inhibition of AMPK attenuated eNOS phosphorylation increased by RUT (Figure 5B). To further assess whether CaMKKβ is required for phosphorylation of eNOS and AMPK, cells were treated with STO-609 prior to treatment with RUT. Pretreatment with STO-609 suppressed RUT-induced the phosphorylation of eNOS and AMPK (Figure 5C). Furthermore, pretreatment with STO-609, Compound C, or KN-62 attenuated the RUT-induced NO generation (Figure 5D). These results suggest that both CaMKII and CaMKKβ/AMPK signaling pathways are crucial upstream kinases of RUT-induced eNOS phosphorylation and NO generation.

### 2.6. Effects of RUT on the Suppression of Monocyte Adhesion and Expression of Adhesion Molecules in Endothelial Cells

NO generation through eNOS activity suppresses the expression of adhesion molecules, which are characteristic of endothelial dysfunction, thereby protecting vascular diseases by suppressing the inflammatory response [23]. To determine the effect of eNOS activity increased by RUT on adhesion molecular expression, endothelial cells were treated with 10 μM RUT for 1 h and additional stimulated with 10 ng/mL tumor necrosis factor-α (TNF-α) for 12 h. RUT suppressed TNF-α-induced the adhesion of THP-1 monocytes to endothelial cells and expression of intercellular adhesion molecule-1 (ICAM-1) and vascular cell adhesion molecule-1 (VCAM-1) in a concentration-dependent manner (Figure 6A,B). Interestingly, pretreatment with 100 μM L-NAME and 10 μM SB366791 significantly reversed inhibitory effects on ICAM-1 and VCAM-1 expression of RUT (Figure 6C). These findings indicated that RUT attenuates TNF-α-induced the adhesion molecules expression by eNOS activity and NO generation.

### 2.7. Effects of RUT on the Inhibition of NF-κB p65 Signaling Pathway Induced by TNF-α in Endothelial Cells

The transcription factor nuclear factor-κB (NF-κB) composed of p65 and p50 subunits is an important factor in the inflammatory response in endothelial cells, and eNOS-derived NO generation is closely related to anti-inflammatory effects [24,25]. To assess the association between the inflammatory signaling pathway and eNOS activation in endothelial cells under RUT treatment, we examined the effect of RUT on TNF-α-induced phosphorylation and translocation of NF-κB p65 in endothelial cells. As shown in Figure 7A, RUT substantially inhibited TNF-α-induced NF-κB p65 phosphorylation. In addition, pretreatment with 100 μM L-NAME and 10 μM SB markedly reversed the inhibitory effect of RUT on the phosphorylation and nucleus protein levels of NF-κB p65 (Figure 7B,C). These results suggest that RUT may protect endothelial function by suppressing the excessive inflammatory response.

## 3. Discussion

Endothelial dysfunction causes cardiovascular diseases, including atherosclerosis and hypertension, through an excessive inflammatory reaction by increasing the adhesion and permeability of immune cells [1,26]. NO synthesized by eNOS in endothelial cells promotes relaxation of vascular smooth muscle and protects vascular endothelial function [27]. In this study, we demonstrated the effects of RUT on NO generation through eNOS phosphorylation in human endothelial cells and its underlying molecular mechanisms. RUT was shown to induce NO synthesis by increasing eNOS phosphorylation in endothelial cells. Furthermore, eNOS phosphorylation by RUT was induced by increasing intracellular Ca^2+^-dependent CaMKKβ, CaMKII, and AMPK signaling pathways through TRPV1 channels.

The eNOS protein sequence has a binding site of calmodulin (CaM), a calcium-modulated protein located between the N-terminal oxygenase domain and the C-terminal reductase domain [28]. Several studies have reported that agonists enhance NO synthesis by CaMKII-dependent eNOS phosphorylation by increasing intracellular Ca^2+^ levels in endothelial cells [5,19]. We found that RUT increased the intracellular calcium levels and the activity of Ca^2+^/CaM-dependent kinases, such as CaMKII and CaMKKβ. In addition, KN-62, a CaMKII inhibitor, suppressed RUT-induced NO production and eNOS phosphorylation, suggesting that RUT increases eNOS phosphorylation through the CaM/CaMKII signaling pathway. Similar to our results, it has been reported that phosphorylation of CaMKII by Ca^2+^ protects endothelial function via eNOS phosphorylation and NO synthesis [29]. CaMKKβ activation is involved in eNOS phosphorylation by inducing the AMPK signaling pathway [30]. CaMKKβ activity induced by high intracellular Ca^2+^ levels was reported to promote NO synthesis and eNOS activity via the AMPK signaling pathway and thus protect against endothelial cell dysfunction [30,31]. Our results showed that RUT significantly increased AMPK phosphorylation, and AMPK inhibition by Compound C attenuated RUT-induced NO generation and eNOS phosphorylation. Furthermore, inhibition of CaMKKβ suppressed RUT-induced NO generation, as well as phosphorylation of AMPK and eNOS. A recent study has reported that the CaMKKβ/AMPK signaling pathway improved diabetes-induced endothelial dysfunction in mouse aorta by inducing eNOS phosphorylation [7]. These results indicated that RUT increased eNOS phosphorylation through the Ca^2+^/CaMKII and Ca^2+^/CaMKKβ/AMPK signaling pathway in endothelial cells.

TRP channels, which are highly expressed in the endothelial cell membrane, are closely involved in endothelial function by regulating intracellular Ca^2+^ influx and cell membrane potential. Furthermore, dysfunction of endothelial TRP channels results in cardiovascular disease through excessive oxidative damage, reduced NO availability, and disruption of the endothelial barrier [32]. A derivative of RUT was reported to show a vasodilatory effect by increasing eNOS expression through TRPV1 activation in human aortic endothelial cells, as well as an anti-inflammatory via suppression of TNF-α and cyclooxygenase-2 in lipopolysaccharide-stimulated macrophages [33]. In a previous study, we demonstrated that RUT could increase the intracellular Ca^2+^ level through TRPV1 channels [16]. Consistent with these results, in the present study, TRPV1 blockade inhibited the RUT-induced increase in intracellular Ca^2+^ level and eNOS phosphorylation in human endothelial cells. These results indicated that TRPV1 plays an important role in eNOS phosphorylation by RUT in endothelial cells.

Endothelial dysfunction is accompanied by a change in the shape of endothelial cells, increasing cell–cell monolayer permeability and expression of adhesion molecules, leading to excessive inflammatory response and invasion of leukocytes [34,35]. Our results showed that RUT suppressed the high expression of ICAM-1 and VCAM-1 induced by TNF-α in human endothelial cells. Moreover, RUT reduced the adhesion of monocytes to the endothelium by TNF-α treatment. Several studies have demonstrated that NO generation by eNOS activity exerted a protective effect on the adhesion of the leukocyte to endothelium by decreasing the expression of adhesion molecules and pro-inflammatory activity [36,37]. Similarly, our results showed that the inhibition of eNOS and TRPV1 reversed the inhibitory effect of RUT on TNF-α-induced expression of ICAM-1 and VCAM-1 expression in human endothelial cells. Furthermore, RUT inhibits the phosphorylation and nucleus protein levels of NF-κB p65 induced by TNF-α through NO generation via the TRPV1 channel. According to recent studies, TRPV1 activation is closely related to the suppression of excessive inflammatory response within the endothelium [33]. These results are consistent with a previous study that the activity of TRPV1 channels attenuated the inflammatory response by increasing the activation of the eNOS/NO pathway in endothelial cells [11]. These results suggest that RUT protects against endothelial dysfunction and excessive inflammatory response by increasing eNOS and NO synthesis through TRPV1 in endothelial cells.

## 4. Materials and Methods

### 4.1. Cell Culture and Treatment

Human endothelial EA. hy926 cells were purchased from the American Type Culture Collection (Manassas, VA, USA). The cells were cultured in Dulbecco’s modified Eagle’s medium (DMEM) supplemented with 10% fetal bovine serum (FBS), 1% penicillin, and streptomycin (Welgene, Gyeonwgsan, Korea). The cells were incubated in DMEM containing 10% FBS at 37 °C in a humidified incubator with 5% CO_2_. RUT was prepared by dissolving in dimethylsulfoxide (DMSO) and adding to the medium, with the final concentration of DMSO not exceeding 0.1%.

### 4.2. MTT and LDH Assays

3-(4,5-Dimethylthiazol-2-yl)-2,5-diphenyltetrazolium bromide (MTT) reduction and lactate dehydrogenase (LDH) assays were used to investigate the toxicity of RUT against cells. MTT was purchased from Sigma-Aldrich (St. Louis, MO, USA), and an LDH release kit was purchased from Roche Applied Science (Indianapolis, IN, USA). Cells were seeded in 48-well plates (5 × 10^4^ cells/well) and incubated at 37 °C for 24 h. Cells were treated with 1–20 μM RUT, and the plates were incubated at 37 °C for 24 h. A BioTek Synergy HT microplate reader (BioTek Instruments, Winooski, VT, USA) was used to determine the absorbance at 550 nm (MTT) and 490 nm (LDH). Calculations of cell viability (%) and cytotoxicity (%) were based on the absorbance of RUT-treated cells relative to the absorbance of vehicle (DMSO 0.1%)-treated cells.

### 4.3. Western Blotting Analysis

Endothelial cells were lysed in CETi lysis buffer (TransLab, Daejeon, Korea) on ice for 30 min and centrifuged at 13,000 rpm for 15 min. Supernatants were collected, and protein concentrations were determined at 595 nm using a protein assay kit (Pro-Measure, iNtRON Biotechnology, Seongnam, Gyeonggi, Korea). Equal amounts of total cellular proteins were boiled for 5 min and subjected to 10% sodium dodecyl sulfate polyacrylamide gel electrophoresis (SDS-PAGE). The proteins were then transferred onto nitrocellulose membranes and incubated with primary antibodies followed by incubation with anti-mouse or anti-rabbit secondary antibody as appropriate. Finally, the densities of protein bands were measured using an enhanced Hisol ECL Plus Detection Kit (BioFact, Daejeon, Korea). Antibodies against p-eNOS, p-CaMKII, p-AMPK, and p-CaMKKβ, as well as anti-mouse and anti-rabbit IgG antibodies, were purchased from Cell Signaling Technology (Beverly, MA, USA). An antibody against β-actin was purchased from Santa Cruz Biotechnology (Santa Cruz, CA, USA).

### 4.4. Measurement of NO Production

The generation of NO was detected using the NO-specific fluorescent dye DAF-2DA (Calbiochem) as described previously [9]. The cells were treated with RUT as indicated in the figure legends. The absorbance of culture media was measured at 495/515 nm using a BioTek Synergy HT microplate reader (BioTek Instruments, Winooski, VT, USA). The cells were fixed in 5% paraformaldehyde for 5 min at 4 °C and visualized using an EVOS fluorescence microscope (Life Technologies, Carlsbad, CA, USA).

### 4.5. Ca^2+^ Measurement

Intracellular calcium levels were evaluated using Fluo-4AM in accordance with the manufacturer’s instructions as described previously [9]. The cells were treated with RUT, as indicated in the figure legends. Fluo-4AM was excited at a wavelength of 488 nm, and emission was monitored at 512 nm. Fluorescence images of the selected cells were captured using an EVOS fluorescence microscope (Life Technologies, Carlsbad, CA, USA).

### 4.6. Monocyte Adhesion

Endothelial cells were seeded in 48-well plates (1 × 10^5^ cells/well) and pre-incubated with RUT for 3 h, and then with 10 ng/mL TNF-α for 12 h. THP-1 cells were fluorescence-labeled using calcein-AM (100 μM, 1 × 10^6^ cells). Endothelial cells and THP-1 cells were co-incubated for 1 h. Non-adherent cells were removed, and the wells were washed twice with PBS. The cells were captured using an EVOS fluorescence microscope.

### 4.7. Statistical Analysis

The experimental data are reported as means ± standard deviation (SD) of at least three independent experiments. One-way analysis of variance (ANOVA) was used to determine the significance of differences between treatment groups. The Newman–Keuls test was used to compare multiple groups. In all analyses, *p* < 0.01 was taken to indicate statistical significance.

## 5. Conclusions

This study investigated the effects of RUT on NO synthesis via eNOS phosphorylation in endothelial cells and the underlying molecular mechanisms. RUT induces increases in intracellular Ca^2+^ levels in endothelial cells via the entry of extracellular Ca^2+^ through TRPV1 channels. High intracellular Ca^2+^ levels stimulate the phosphorylation of CaMKII and CaMKKβ, which are upstream regulators of AMPK. The activation of CaMKII and AMPK by RUT increases eNOS phosphorylation and NO generation. Furthermore, the increased activity of eNOS and NO synthesis by RUT-suppressed TNF-α-induced inflammatory signaling pathways and the expression of adhesion molecules. These findings suggest that the TPRV1 channel not only plays a protective role in the inflammatory response of endothelial cells under TNF-α-induced inflammatory conditions but also has an essential role in RUT-conferred protection of endothelial cells.

## Figures and Tables

**Figure 1 ijms-22-09407-f001:**
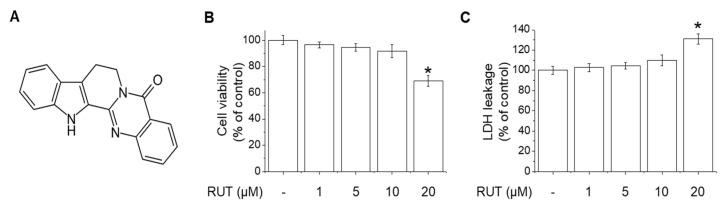
Effects of RUT on viability of, and cytotoxicity in, endothelial cells. (**A**) Chemical structure of RUT. (**B**) Cell viability was assessed by MTT assay. (**C**) Cytotoxicity was assessed by LDH assay. * Significantly different from control at *p* < 0.01.

**Figure 2 ijms-22-09407-f002:**
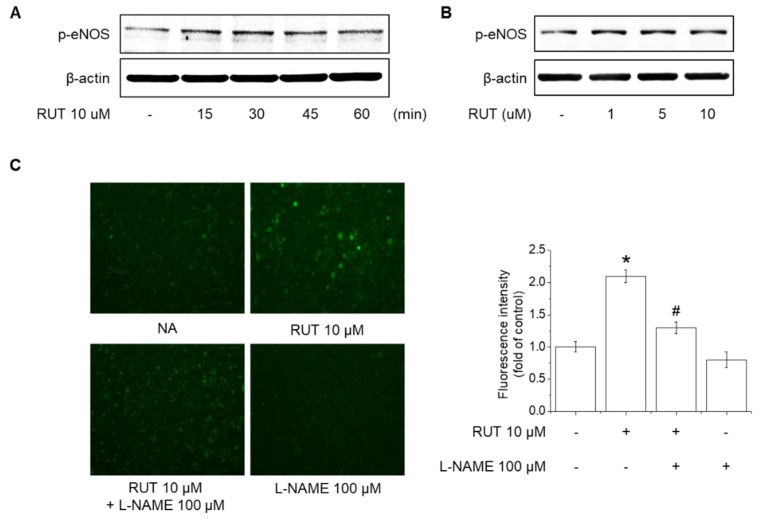
Effects of RUT on eNOS phosphorylation and NO generation. Cells were treated with (**A**) 10 µM RUT for 10–60 min or (**B**) 1–10 µM RUT for 30 min and assessed by Western blotting. (**C**) Cells were pretreated with 100 µM L-NAME for 1 h and then 10 µM RUT for 90 min. NO generation was detected using DAF-2DA. * Significantly different from control at *p* < 0.01. ^#^ Significantly different from RUT-treated cells at *p* < 0.01.

**Figure 3 ijms-22-09407-f003:**
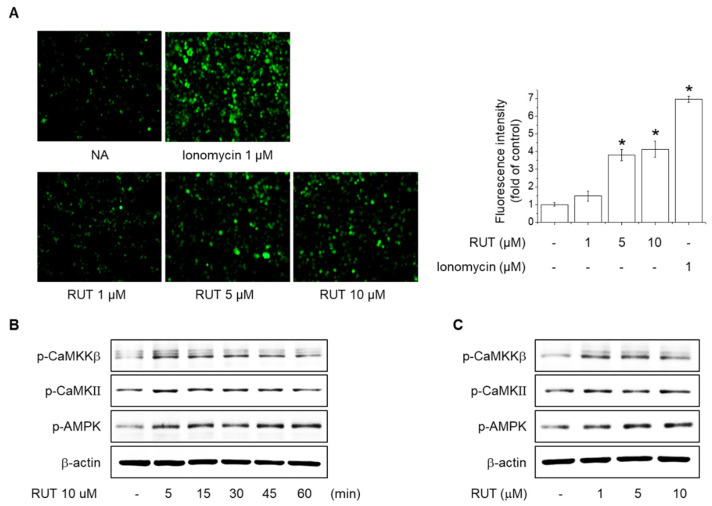
Effects of RUT on levels of intracellular calcium and phosphorylation of CaMKII, CaMKKβ, and AMPK. (**A**) Intracellular calcium was detected using the fluorescent calcium indicator, Fluo-4-AM. Fluo-4-AM-treated cells were stimulated with 1–10 µM RUT for 5 min. * Significantly different from control at *p* < 0.01. Cells were treated with (**B**) 10 µM RUT for 5–60 min or (**C**) 1–10 µM RUT for 60 min and assessed by Western blotting.

**Figure 4 ijms-22-09407-f004:**
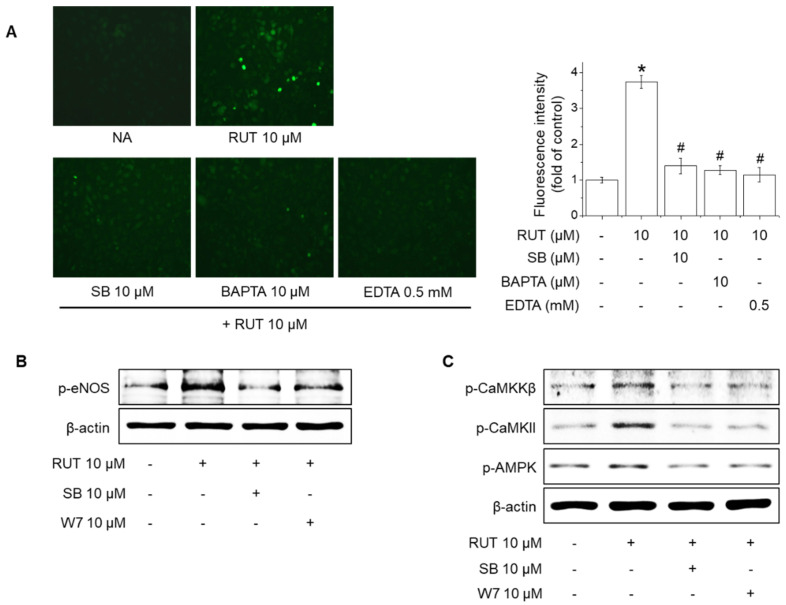
Effects of TRPV1 blockade on levels of intracellular calcium and phosphorylation of CaMKKβ, CaMKII, AMPK, and eNOS in response to RUT. (**A**) The effects of RUT on intracellular calcium influx in endothelial cells are mediated via TRPV1 channels. Cells were pretreated with 10 μM SB366791 (SB), 10 μM BAPTA-AM (BAPTA), or 10 μM EDTA for 30 min before Fluo-4-AM treatment. Then, the Fluo-4-AM-treated cells were stimulated with 10 µM RUT for 5 min. * Significantly different from control at *p* < 0.01. ^#^ Significantly different from RUT-treated cells at *p* < 0.01. (**B**) Effect of TRPV1 and calmodulin inhibition on eNOS phosphorylation by RUT. Cells were treated with 10 μM SB and 10 μM W7 for 30 min and then stimulated with 10 μM RUT for 30 min. (**C**) Cells were treated with 10 μM SB and 10 μM W7 for 30 min and then stimulated with 10 μM RUT for 15 min. Western blotting was performed to assess phosphorylation of CaMKKβ, CaMKII, AMPK, and eNOS.

**Figure 5 ijms-22-09407-f005:**
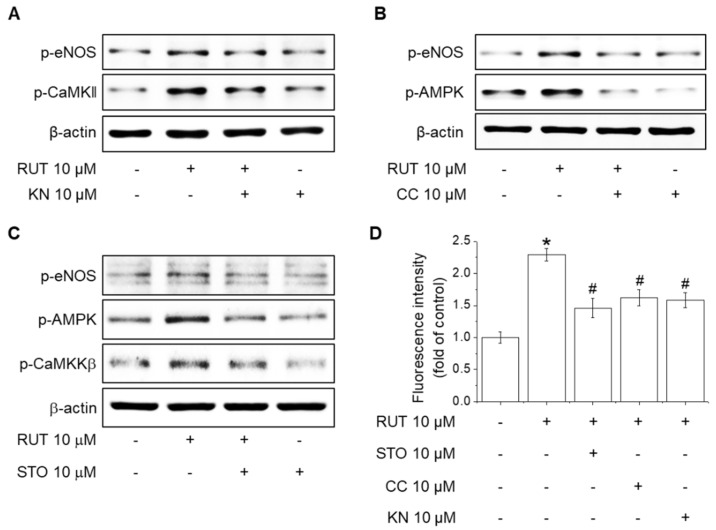
Roles of CaMKII activation and the CaMKKβ/AMPK signaling pathway in RUT-induced eNOS activation and NO generation. (**A**) Effects of CaMKII inhibition on eNOS phosphorylation induced by RUT. Cells were pretreated with 10 μM KN-62 (KN) for 30 min and then incubated with 10 μM RUT for a further 30 min. (**B**) Effects of AMPK inhibition on eNOS phosphorylation induced by RUT. Cells were pretreated with 10 μM Compound C (CC) for 30 min and then incubated with 10 μM RUT for a further 30 min. (**C**) Effects of CaMKKβ inhibition on eNOS phosphorylation induced by RUT. Cells were pretreated with 10 μM STP-609 (STO) for 30 min and then incubated with 10 μM RUT for a further 30 min. Western blotting was performed to assess phosphorylation of CaMKKβ, CaMKII, AMPK, and eNOS. (**D**) Effects of CaMKII, AMPK, and CaMKKβ inhibition on increased NO generation by RUT. Cells were pretreated with 10 μM STO, CC, or KN for 30 min prior to incubation with DAF-2DA for 30 min and treatment with 10 μM RUT for a further 90 min. * Significantly different from control at *p* < 0.01. ^#^ Significantly different from RUT-treated cells at *p* < 0.01.

**Figure 6 ijms-22-09407-f006:**
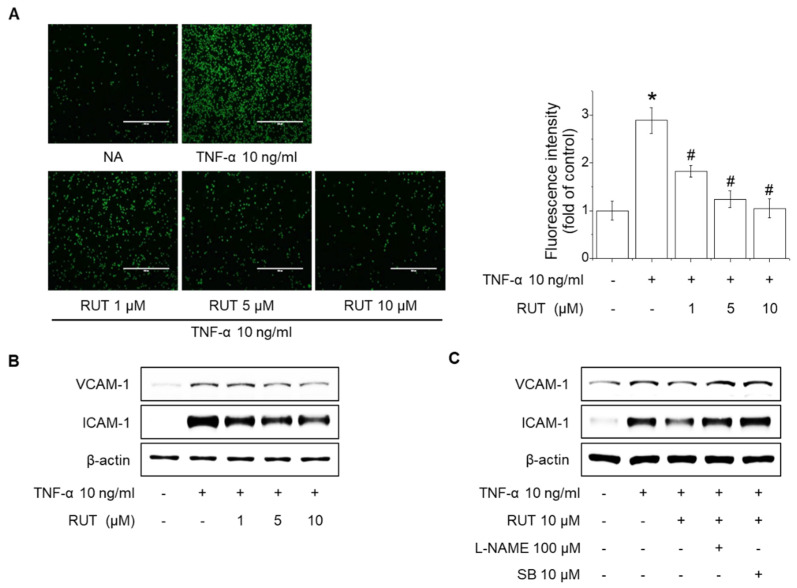
Inhibitory effect of RUT on the expression of adhesion molecules and adhesion of monocytes to endothelial cells. (**A**) Endothelial cells were treated with 1–10 μM RUT for 3 h, and TNF-α was added at 10 ng/mL for an additional 12 h. Endothelial cells were co-cultured with THP-1 monocytes for 1 h, and the adherence of endothelial cells to monocytes was assessed by fluorescence microscopy. * Significantly different from control at *p* < 0.01. ^#^ Significantly different from RUT-treated cells at *p* < 0.01. (**B**) Cells were treated with various concentrations (1–10 μM) of RUT for 3 h, followed by incubation with 10 ng/mL TNF-α for 12 h. (**C**) Cells were pretreated with 100 μM L-NAME or 10 μM SB366791 (SB) for 1 h, and then treated with 10 μM RUT for an additional 3 h, followed by incubation with 10 ng/mL TNF-α for 12 h. Western blotting was performed to assess expression of ICAM-1 and VCAM-1.

**Figure 7 ijms-22-09407-f007:**
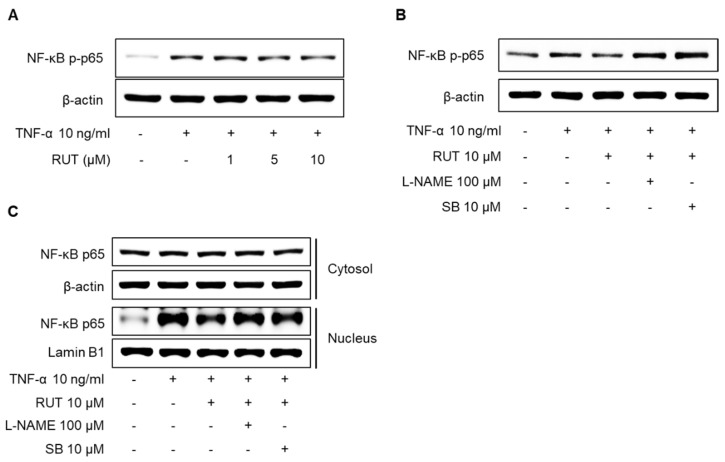
Inhibitory effect of RUT on the TNF-α-induced NF-κB signaling pathway in human endothelial cells. (**A**) Cells were treated with various concentrations (1–10 μM) of RUT for 1 h, followed by incubation with 10 ng/mL TNF-α for 30 min. (**B**) Cells were pretreated with 100 μM L-NAME or 10 μM SB366791 (SB) for 1 h, and then treated with 10 μM RUT for an additional 1 h, followed by incubation with 10 ng/mL TNF-α for 30 min. (**C**) Cells were pretreated with 10 μM RUT for 1 h, followed by incubation with 10 ng/mL TNF-α for 3 h. Cell lysates were analyzed by Western blotting using specific antibodies.

## Data Availability

The data presented in this study are available on request from the corresponding author.

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
