# Peer review of "Rutaecarpine Increases Nitric Oxide Synthesis via eNOS Phosphorylation by TRPV1-Dependent CaMKII and CaMKKβ/AMPK Signaling Pathway in Human Endothelial Cells"

_ijms, 2021, doi:10.3390/ijms22179407_

Round 1

Reviewer 1 Report

The results are clearly presented. The most interesting effect connects with the  increasing of activity of eNOS and NO synthesis by RUT suppressed TNF-α- induced inflammatory signaling pathways and expression of adhesion molecules. The methods used are adequate. Their description is not always easy to follow. Conclusions are the supported by the results and have fundemantal and clinical importance.

Author Response

Thank you for the great comment. We have checked the manuscripts, and some sentences have been corrected for easier understanding.

Reviewer 2 Report

The paper entitled "Rutaecarpine increases nitric oxide synthesis via eNOS phosphorylation by TRPV1-dependent CaMKII and CaMKKβ / AMPK signaling pathway in human endothelial cells" explores the effect of rutaecarpine, a natural bioactive alkaloid, on human endothelial cells and molecular mechanisms of action. In this study it is shown that rutaecarpine treatment promoted nitric oxide generation (with protective role) by increasing endothelial nitric oxide synthase phosphorylation. Moreover, the mechanism of action is described. All chapters are described accordingly and the conclusions are supported by the results. It is a complex study that deserves to be published in IJMS. The only specific comment is to enlarge the figures so that the text can be read.

Author Response

Thank you for the great comment. Following your suggestion, we enlarged the figures so that the text can be read as following.

Reviewer 3 Report

see the attachment

Author Response

Thank you for the great comment. Based on your comments, we have corrected the manuscript as follows.
